# Inactivation of *Aeromonas hydrophila* and *Vibrio parahaemolyticus* by Curcumin-Mediated Photosensitization and Nanobubble-Ultrasonication Approaches

**DOI:** 10.3390/foods9091306

**Published:** 2020-09-16

**Authors:** Shamil Rafeeq, Setareh Shiroodi, Michael H. Schwarz, Nitin Nitin, Reza Ovissipour

**Affiliations:** 1Virginia Seafood Agricultural Research and Extension Center, Virginia Tech, Hampton, VA 23669, USA; shamilrafeeq363@gmail.com (S.R.); shiroodi@vt.edu (S.S.); mschwarz@vt.edu (M.H.S.); 2Center for Coastal Studies (Coastal@VT), Virginia Tech, Blacksburg, VA 24061, USA; 3Department of Food Science and Technology, University of California-Davis, Davis, CA 95616, USA; nnitin@ucdavis.edu; 4Department of Agricultural and Biological Engineering, University of California-Davis, Davis, CA 95616, USA; 5Department of Food Science and Technology, Virginia Tech, Blacksburg, VA 24060, USA

**Keywords:** curcumin, LED, UV-A, nanobubbles, aquatic pathogens, aquaponics

## Abstract

The antimicrobial efficacy of novel photodynamic inactivation and nanobubble technologies was evaluated against *Vibrio parahaemolyticus* and *Aeromonas hydrophila* as two important aquatic microbial pathogens. Photodynamic inactivation results showed that LED (470 nm) and UV-A (400 nm)-activated curcumin caused a complete reduction in *V. parahaemolyticus* at 4 and 22 °C, and a greater than 2 log cfu/mL reduction in *A. hydrophila*, which was curcumin concentration-dependent (*p* < 0.05). Furthermore, the photodynamic approach caused a greater than 6 log cfu/mL *V. parahaemolyticus* reduction and more than 4 log cfu/mL of *A. hydrophila* reduction in aquaponic water samples (*p* < 0.05). Our results with the nanobubble technology showed that the nanobubbles alone did not significantly reduce bacteria (*p* > 0.05). However, a greater than 6 log cfu/mL *A. hydrophila* reduction and a greater than 3 log cfu/mL of *V. parahaemolyticus* reduction were achieved when nanobubble technology was combined with ultrasound (*p* < 0.05). The findings described in this study illustrate the potential of applying photodynamic inactivation and nanobubble–ultrasound antimicrobial approaches as alternative novel methods for inactivating fish and shellfish pathogens.

## 1. Introduction

Fisheries and aquaculture are a growing industry, and seafood consumption has been increased from an average of 9.9 kg per capita in the 1960s, to 20.3 kg per capita in 2017 [1]. Pathogenic *Vibrio* spp., specifically, *Vibrio parahaemolyticus*, are the leading causes of seafood-associated disease in U.S., and 45% of the seafood-borne outbreaks are related to molluscan shellfish [2]. Vibrio spp. are natural inhabitants of estuaries and coastal marine environments. They can be found in water, sediments, and all flora and fauna in coastal environments, including freshly harvested seafood. Another aquatic important microorganism is *Aeromonas hydrophila* which can cause disease in both fish and humans, affecting seafood safety, quality, and causing severe losses for production and marketing [3]. The importance of controlling these pathogenic bacteria is highlighted by the fact that the aquaponics industry is growing globally, and the number of aquaponics producers in the U.S. has continued to grow [4]. Since most fresh produce is consumed raw, the potential for cross-contamination and transfer of pathogenic bacteria from aquaculture water to edible parts of the plants represents a serious risk to public health [5]. Controlling zoonotic fish diseases and foodborne pathogens in recirculating aquaculture systems and aquaponics using antibiotics, chemical sanitizers, and pesticides, is challenging due to the sensitivity of the microbial community in biofilters which oxidize ammonia to nontoxic nitrate, chemical residue concerns in both fish and plants, and regulatory stand points. These challenges motivated researchers to develop novel chemical-free, or bio-based antimicrobial approaches. For example, the application of light-based interventions has emerged recently as an alternative approach to inactivate bacteria [6,7,8]. The photosensitizer curcumin, in combination with light, has been used for inactivation of various pathogenic and spoilage microorganisms [7,8,9,10,11,12]. Exposure of curcumin to light radiation results in photooxidative generation of reactive oxygen species (ROS), which have strong antimicrobial properties [7,8]. Recent reports have documented curcumin-mediated photooxidative DNA damage as a mechanism for bacterial inactivation [7,8,13]. Light emitting diodes (LED) and UV-A are two main safe sources of lights for photodynamic inactivation which have been used to improve sanitation of food products [7,8,14].

Nanobubble technology represents another promising antimicrobial approach that has been recently described. Cavitative collapse of nanobubbles generates reactive oxygen species, as well as a physical insult to microbial cell structures [15,16,17,18]. Due to nanobubbles’ unique properties, this technology has been applied in various areas of advanced science and technology including engineering, medical, agricultural, and food sectors [16,17,18,19,20,21]; for cleaning surfaces [22,23,24]; dental hygiene [25]; wound cleaning [16,17]; removing bacteria from fresh produce [26]; inactivation of norovirus [27]; and removing microbial biofilms [21]. Nanobubbles can exist in both bulk solution and at liquid–solid interfaces, and due to their unique physical properties (nanosize diameter, negative surface charge and Brownian motion), nanobubbles can remain stable for up to 24 h, resulting in a supersaturated bubble phase [15]. However, only a few studies have investigated the antimicrobial properties of nanobubbles alone or in combination with chemicals and other nonthermal processes such as ultrasound [21,26,27]. Thus, to address the potential challenges in water sanitation in Recirculating Aquaculture Systems (RAS) and aquaponics, we propose the use of two technologies including curcumin-mediated photosensitization and nanobubbles with ultrasound to inactivate *V. parahaemolyticus* and *A. hydrophila*. The efficacy of light (LED and UV-A)-activated curcumin at different concentrations and exposure time and temperature against these bacteria was evaluated. Antimicrobial properties of synergistic approaches of nanobubbles–ultrasound were also determined at different exposure times at room temperature.

This study illustrates the potential of light activated food grade antimicrobial materials, such as curcumin, and a novel chemical-free approach combining nanobubbles and ultrasound for water sanitation in intensive RAS and aquaponics systems to reduce reliance on chemical-based approaches.

## 2. Materials and Methods

### 2.1. Bacterial Strains and Inoculum Preparation

Clinically isolated *A. hydrophila* was provided by David Crosby from Virginia State University, and *V. parahaemolyticus*, was isolated from sea water. Frozen stock cultures of the strains were streaked on agar media and incubated at 35 and 37 °C for 24 h, for *V. parahaemolyticus* and *A. hydrophila*, respectively. A loop of these cultures was transferred two successive times into 10 mL tryptic soy broth and incubated at 37 and 35 °C (*V. parahaemolyticus*) for 18 h. A total of 1 mL of the broth culture was pipetted into a 1.5 mL centrifuge tube and centrifuged at 10,483× *g* for 2 min. Supernatant was discarded, and the resultant the pelleted cells were resuspended with 1 mL sterile phosphate buffer saline (PBS), and centrifuged at 10,483× *g* for 2 min. This process was repeated twice, and the pellet from the second wash was resuspended in 1 mL sterile PBS. The population of bacteria in inoculum was approximately 10^9^ CFU/mL.

### 2.2. Preparation of Curcumin Solutions

Different final concentrations of curcumin (2, 10, and 20 mg/L) were prepared by diluting the 20 mM stock solution of curcumin with deionized water. Stock solution was prepared by dissolving 737 mg curcumin in 100 mL of 99% ethanol according to Oliveira et al. [7,8]. These concentrations were selected based our preliminary results and according to Oliveira et al. [7,8].

### 2.3. Light Irradiation of Bacteria

Two light sources were used in this study including UV-A (320–400 nm; 18W; Actinic BL, Philips, Holland) and light-emitting diode (LED) arrays (470 nm; energy density of 3.6 J/cm^2^, AquaBasik, Yescom USA, Inc., City of Industry, CA, USA).

The efficacy of curcumin activated by UV-A and LED lights was studied against *V. parahaemolyticus* and *A. hydrophila*. Samples with curcumin were prepared by adding 5 mL fresh *V. parahaemolyticus* and *A. hydrophila* cell suspensions to 5 mL curcumin solution (2, 10, and 20 mg/L). The final concentration of the cells was 10^6^ cfu/mL. Samples were placed into sterile 6-well clear polystyrene microplates and were treated by UV-A for 5 and 15 min, and LED for 15 and 30 min (based on our preliminary data, no bacterial reduction was observed within 15 min exposure to LED) at 4 and 22 °C. After exposure, *V. parahaemolyticus* and *A. hydrophila* were cultured on TCBS (Thiosulfate Citrate Bile Salts Sucrose) and TSA (Tryptic Soy Agar), and incubated at 35 and 37 °C for 48 h, respectively. Bacteria with curcumin without exposure to lights, and bacteria without curcumin exposed to lights were used as control groups. All the experiments were repeated at least two times in triplicate (*n* = 6). The pH of the water samples was adjusted to 5.6 using 1 N citric acid.

### 2.4. Photodynamic Inactivation in RAS-Aquaponics Water

RAS-aquaponics water sanitation was simulated using water from a seventy liter pilot-scale RAS-aquaponics system (tilapia-pepper, Built by Virginia Seafood AREC). Before conducting the experiment, to ensure that there were no *Vibrio* sp. and *Aeromonas* in the RAS-aquaponics water, 100 mL RAS water was filtered using 0.45 µm, and a filter was placed on TCBS and selective *Aeromonas* medium (RYAN) (CM0833, Oxoid Ltd, Basingstoke, UK), and no bacterial colonies were observed. The pH of the water samples was adjusted to 5.6 using 1 N citric acid. Then, samples were prepared as mentioned previously, by mixing 5 mL of the pH-adjusted aquaponics water with bacterial cell suspensions to obtain 10^6^ cfu/mL and 10 mg/L curcumin concentrations, which were treated with light sources for 5, 10, and 20 min.

### 2.5. Nanobubble Inactivation of Bacteria

The synergistic antimicrobial activity of nanobubbles and ultrasound was studied against *V. parahaemolyticus* and *A. hydrophila*. A nanobubble solution was provided using deionized water and pure oxygen gas using a Moleaer 25 L nanobubble generator (Moleaer Inc., Torrance, CA, USA). Our preliminary experiments showed that nanobubbles produced by pure oxygen showed stronger antibiofilm and antimicrobial properties as compared to nanobubbles generated by pure carbon dioxide or air. A total of 1 mL of the previously washed bacterial cell suspensions was added to 9 mL deionized water containing nanobubbles. The solutions were exposed to ultrasound for 5, 10, and 15 min. Bacteria in PBS with ultrasound and bacteria in nanobubbles without ultrasound were used as control.

### 2.6. Data Analysis

Each experiment was conducted a minimum of three times to ensure reproducibility. The results were expressed as the mean of the replicates ± standard deviation. The significance of differences among the biofilm removal treatments was determined using one-way analysis of variance (ANOVA), and differences were considered significant at *p* < 0.05. The normality of the data was determined using one sample Kolmogorov–Smirnov test using JMP^®^ Pro 15.0.0 (SAS Institute Inc., Cary, NC, USA).

## 3. Results and Discussion

### 3.1. Photodynamic Bacterial Inactivation

#### 3.1.1. Impact of Curcumin Concentration and Light Wavelength

Figure 1 illustrates the impact of light (UV-A) in combination with curcumin at various concentrations on *V. parahaemolyticus* and *A. hydrophila* inactivation within 15 min with the limit of detection below 0.5 log cfu/mL (**). UV-A activated curcumin with concentrations of 1, 5, and 10 mg/L reduced from 2 to 5 log cfu/mL of *V. parahaemolyticus*, and from 1 to 4.5 log cfu/mL of *A. hydrophila*, indicating that UV-A activated curcumin at low concentrations has strong antimicrobial properties, and by increasing the curcumin concentration from 1 to 10 mg/L, the antimicrobial activity increased significantly. Additional controls including bacteria with curcumin and no light and bacteria without curcumin and light, with no reduction, were applied (data not shown).

Bacteria were also exposed to LED light (470 nm) for 15 and 30 min with curcumin (our preliminary experiments showed no significant reduction within 10 min exposure). The results indicated that increasing the curcumin concentration from 1 to 10 mg/L resulted in increasing the efficacy of the combination approach to inactivate bacteria. Curcumin concentration demonstrated a strong impact on the reduction in *V. parahaemolyticus* and *A. hydrophila* regardless of the light source. Similar results were previously reported by other researchers. Oliveira et al. [7] reported an increase in the inactivation of *Listeria innocua* and *E. coli* O157:H7 by increasing curcumin concentrations and activating by UV-A, and Wu et al. [12] demonstrated an increase in *V. parahaemolyticus* inactivation by increasing curcumin concentrations (5 to 20 µM) and LED.

Overall, the results indicated that *V. parahaemolyticus* was less resistant to light-activated curcumin compared to *A. hydrophila*. A higher reduction was also observed for *V. harveyi* compared to *A. salmonicida* [28], and *V. parahaemolyticus* exhibited a faster rate of inactivation compared to *Staphylococcus aureus* and *Lactobacillus plantarum* in the presence of 405 as well as 470 nm LED illumination [29]. Gram-negative bacteria (*E. coli* O157:H7) are more resistant to light-activated curcumin, compared to Gram-positive bacteria (*L. innocua*) [7]), which is mainly because of the Gram-positive bacteria outer wall structure permeability for curcumin [30]. The outer-wall structure of Gram-positive bacteria contains up to 100 peptidoglycan layers, which display a relatively high degree of porosity, which is permeable to molecules such as curcumin [30]. However, on the contrary, a higher sensitivity of Gram-negative bacteria compared to Gram-positive bacteria has also been reported [29,31]. Furthermore, our results indicated that UV-A 400 nm caused a higher bacterial reduction compared to LED 470 nm. After 5 min of light treatment, a greater than 5 log cfu/mL of *V. parahaemolyticus* and a greater than 4 log cfu/mL reduction in *A. hydrophila* were achieved by UV-A light. However, LED 470 nm only reduced 3 and 3.5 log cfu/mL of *V. parahaemolyticus* and *A. hydrophila*, respectively. Similar results have been previously observed by other researchers, where at an equal radiant energy dosage of 1100 J/cm^2^, the 395 nm LED treatment was more effective with a 2.48 log cfu/g reduction, than the 455 nm with a 1.6 log cfu/g reduction in *Salmonella* in wheat flour [14]. Kumar et al. [29] also reported low antimicrobial properties of 460 nm compared to 405 nm, regardless of bacterial species and illumination temperature. This can be explained by the fact that UV-A (400 nm) can generate ROS and directly cause microbial cell damage compared to visible light [29], and UV-A light may excite curcumin molecules compared to LED 470 nm [7]. UV-A has been used to activate curcumin for treating *E. coli* and *L. innocua* in water and fresh produce [7,8]. Moreover, mainly LED or visible light were used in combination with photosensitizers for inactivating bacteria including *Vibrio* sp. and *Aeromonas* sp. [12,28,29,32,33].

The results of these experiments illustrate the potential of using the photodynamic inactivation approach for food production systems for which using antibiotics, therapeutics, and pesticides is restricted such as intensive indoor aquaculture systems, aquaponics, and the oyster industry.

#### 3.1.2. Impact of Temperature

Similar to the room temperature (22 °C) experiment, curcumin and UV-A light together caused a more than 5 log cfu/mL reduction in *V. parahaemolyticus* and a more than 4 log cfu/mL reduction in *A. hydrophila* at 4 °C (Figure 1a–d). Compared to UV-A treatment, LED 470 nm only caused a 4.5 and 2.5 log cfu/mL reduction in *V. parahaemolyticus* and *A. hydrophila*, respectively at 4 °C (Figure 2a–d). Temperature control is a critical food safety and quality step during seafood harvesting (particularly oysters) and processing. It has been reported that antimicrobial properties of UV-A light-activated bio-based compounds (e.g., gallic acid and lactic acid) were inhibited at refrigerated temperatures against *E. coli* O157:H7 and *L. innocua* [6,7,8], which is due to the lower bacterial metabolism rates at a lower temperature. Other studies also indicated that bacterial inactivation in the presence of antimicrobial compounds increases with increasing temperature [29] due to the higher metabolism rate and cellular activity [7,8,29]. Due to the higher degree of cellular activity and thus more metabolic burden at a higher temperature, bacteria are more susceptible to antimicrobial agents. Several possible reasons could be attributed to the ability of light-activated curcumin at 4 °C to inactivate bacteria, including: light activation of curcumin is independent from bacterial metabolism [7]; increased proportion of unsaturated fatty acids in bacterial membrane at lower temperatures [29,31]; entering some bacterial species such as *V. parahaemolyticus* to a viable but not culturable (VBNC) state at lower temperatures [29]. Thus, the antimicrobial properties of light-activated curcumin at refrigerated temperatures indicate that photodynamic inactivation using curcumin could be a promising approach for sanitizing seafood, live organisms (shellfish), and water to maximize the bacterial inactivation and shelf life of the products.

#### 3.1.3. Bacterial Inactivation in RAS-Aquaponics Water

The antibacterial properties of curcumin activated by UV-A light against *V. parahaemolyticus* and *A. hydrophila* was evaluated in aquaponics water with 6 mg/L total suspended solid (TSS) to best mimic the application of this approach in the aquaculture or aquaponics industry. Figure 3 shows the bactericidal properties of the combination of curcumin and UV-A light against *V. parahaemolyticus* and *A. hydrophila* within 20 min exposure. As represented in Figure 3a, TSS reduced the efficacy of the UV-A and curcumin combination. In the control group, which was deionized water, a more than 6 log cfu/mL inactivation of *V. parahaemolyticus* was achieved within 5 min, while in aquaponics water even after 20 min, the total reduction was 5.2 log cfu/mL. In addition, the UV-A light–curcumin combination approach reduced 5.5 and 4.1 cfu/mL of *A. hydrophila*, in deionized water and aquaponics, respectively. Overall, a lower bacterial inactivation was observed in aquaponics water with 6 mg/L TSS. The efficacy of antimicrobial agent is highly depending on the presence of organic load [7,8]. Organic materials can significantly reduce the antimicrobial activity by reacting with sanitizers, or protecting microorganisms. However, in our study, a more than 4 log cfu/mL bacterial reduction was achieved during the photodynamic inactivation using curcumin (10 mg/L) and UV-A light in real aquaponics model with TSS, suggesting that photodynamic inactivation using curcumin and UV-A light has strong potential for water sanitation in aquaculture and aquaponics systems, and the efficacy of this approach was not significantly influenced by TSS.

### 3.2. Nanobubble and Ultrasound Inactivation

Figure 4 illustrates the bacterial count in cell suspension after exposure to nanobubbles, ultrasound, and nanobubbles+ultrasound for 5 to 15 min. The results showed that ultrasound and nanobubbles solutions alone cannot significantly reduce bacteria (*p* > 0.05). However, the combination of nanobubbles and ultrasound induced a more than 3 and 6 log cfu/mL reduction in *V. parahaemolyticus* and *A. hydrophila*, respectively. The results of *V. parahaemolyticus* showed that the bacterial reduction was independent of the exposure time, while more *A. hydrophila* reduction was observed by increasing the exposure time.

Ultrasound successfully reduced the bacterial population when it was applied with chemical sanitizers [34,35]. Ultrasound alone and in combination with UV-C was applied for inactivating heterotrophic bacteria in intensive aquaculture system which did not result in a significant reduction [36].

Only few studies showed antimicrobial and antibiofilm properties of nanobubbles in combination with chemical sanitizers [16,17,21,25,26]. In our previous study, we showed antibiofilm properties on nanobubbles alone and in combination with chemical sanitizers [21].

Antibiofilm and antimicrobial properties of nanobubbles are strongly correlated to the physical properties of the nanobubbles (including high gas transfer capacity), generation of free radicals including (OH^•^), high energy release by bursting and collapsing of bubbles, and the potential generation of reactive oxygen species (ROS) [21,24,37]. Due to the high internal pressure in nanobubbles, when they burst on the surface of bacteria because of the ultrasound, they release high surface energy, allowing the conversion of O_2_ to ROS, and cause surface cavitation resulting in bacterial inactivation [18]. Furthermore, hydroxyl radicals can be generated in the water as a result of nanobubble collapse [37]. Hydroxyl radical generation and shock waves from the collapse of small cavities are other possible mechanisms for antimicrobial properties of nanobubbles [16,17]. In our previous study, using Raman spectroscopy, we also showed that nanobubbles cause bacterial DNA degradation, protein oxidation, and cell membrane damage [21].

In an intensive aquaculture system including RAS and aquaponics, appropriate microbial control measures can be achieved via water disinfection using chemical or physical (UV-C) approaches. However, chemical application in RAS and aquaponics systems is limited due to many reasons including: lack of approved therapeutants and chemicals; concerns about compromising fish health, worker health, and environmental safety; chemical residue in fish and plants; risk of impairing microbial communities in biofilters which are oxidizing ammonia, and the functionality of the system strongly depends on them [36,38]. Thus, light–curcumin and nanobubbles–ultrasound combinations could be interesting topics for the fresh produce, aquaculture, and algae mass production systems in the photobioreactors, hydroponics, and aquaponics industries as alternative approaches to treat recycling water. Develop a photo- or nanobubble-reactor may provide a better approach for water treatment in RAS and aquaponics which will solve the permit requirement issues as well.

## 4. Conclusions

UV-A and visible LED in combination with food-grade curcumin can rapidly inactivate aquatic pathogens in water. High bacterial inactivation using curcumin at room and refrigerator temperatures offer a new possible approach for seafood sanitation. Furthermore, by considering the fact that the studied concentrations (1, 5, and 10 mg/L) are safe for fish [39], and the UV-A and curcumin combination reduced more than 4 log cfu/mL of bacteria in aquaponics water, this approach may be applied for real intensive aquaculture, and aquaponics system. Antimicrobial properties of nanobubbles and ultrasound against aquatic pathogens provide a new antimicrobial approach for the first time for water sanitation.

In summary, both light–curcumin and nanobubbles–ultrasound approaches may provide novel alternative sanitation approaches for sensitive systems including aquaculture, aquaponics, hydroponics, algae mass production in reactors, and cell-culture systems.

## Figures and Tables

**Figure 1 foods-09-01306-f001:**
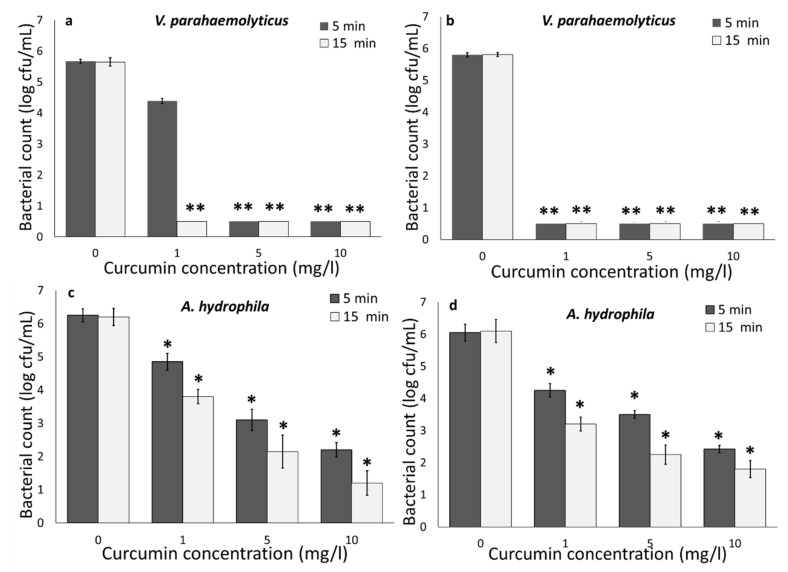
Effect of curcumin concentrations on the inactivation of *V. parahaemolyticus* at (**a**) 22 and (**b**) 4 °C, and *A. hydrophila* at (**c**) 22 and (**d**) 4 °C, using UV-A and curcumin. Statistical difference was determined based on *p* < 0.05 (*). Samples with two asterisks (**) were below the limit of detection (0.5 log cfu/mL).

**Figure 2 foods-09-01306-f002:**
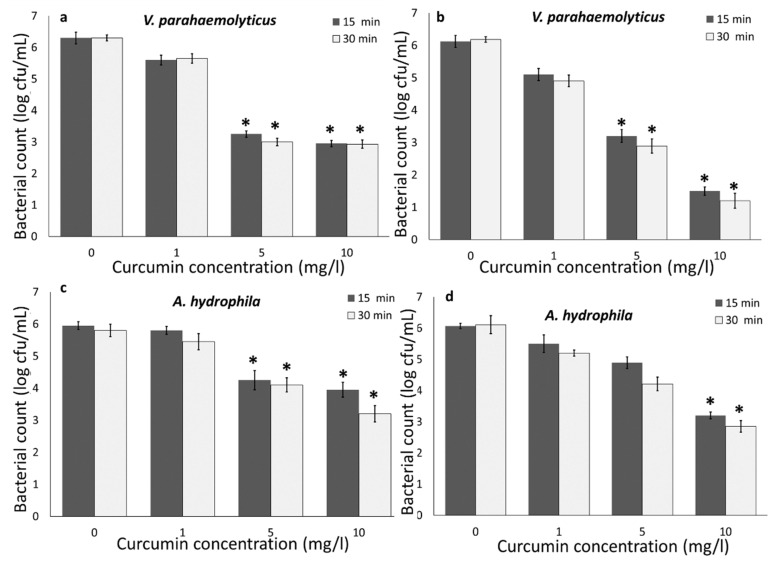
Effect of curcumin concentrations on the inactivation of *V. parahaemolyticus* at (**a**) 22 and (**b**) 4 °C, and *A. hydrophila* at (**c**) 22 and (**d**) 4 °C, using LED and curcumin. Statistical difference was determined based on *p* < 0.05 (*).

**Figure 3 foods-09-01306-f003:**
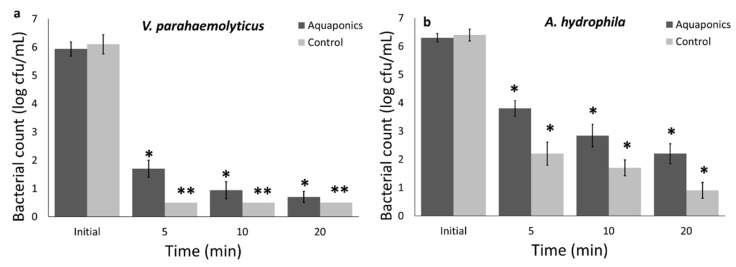
(**a**) *V. parahaemolyticus* and (**b**) *A. hydrophila* inactivation in PBS and aquaponics water with 6 mg/L total suspended solid (TSS), using combined UV-A and curcumin antimicrobial approach. Statistical difference was determined based on *p* < 0.05 (*). Samples with two asterisks (**) were below the limit of detection (0.5 log cfu/mL).

**Figure 4 foods-09-01306-f004:**
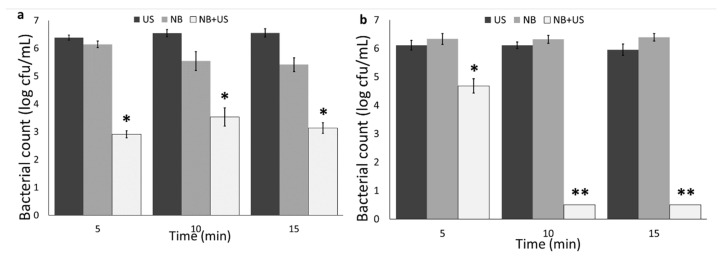
(**a**) *V. parahaemolyticus* and (**b**) *A. hydrophila* count in nanobubbles (NB), ultrasound (US), and nanobubbles + ultrasound (NB + US) at different exposure times. Statistical difference was determined based on *p* < 0.05 (*). Samples with two asterisks (**) were below the limit of detection (0.5 log cfu/mL).

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
