# Peer review of "Inactivation of Aeromonas hydrophila and Vibrio parahaemolyticus by Curcumin-Mediated Photosensitization and Nanobubble-Ultrasonication Approaches"

_foods, 2020, doi:10.3390/foods9091306_

Round 1
Reviewer 1 Report
Review Manuscript foods-889787
REVIEWER COMMENTS TO AUTHORS
Title: Inactivation of Aeromonas hydrophila and Vibrio parahaemolyticus by curcumin-mediated photosensitization and nanobubbles-ultrasonication approaches
Presentation, overall organization and format
The title is clear and adequate.
The English is correct, the manuscript reads easily.
Originality, scientific quality, relevance to the field of this journal
The manuscript is original because the authors study the potential of light activated food grade antimicrobial materials such as curcumin, enhancing the antimicrobial properties of this natural antimicrobial. In addition novel technologies are investigated, nanobubbles and/or ultrasound, for water sanitation in intensive aquaponics systems to reduce therapeutics and chemicals applications.
- Introduction
Line 48 change “photosensitizes” by “photosensitizers, which”
Line 52, Recently? Reference 13, year 2009, more than ten years?
Line 54 “for sanitation” change by “to improve sanitation of food products”
Line 60 “inactivating microbial biofilms” change by “removing microbial biofilms”
Line 65 define RAS
Line 66 “including photosensitizer” change by “curcumin-mediated photosensitization” and remove “(curcumin and LED and or UV-A)”
- Materials and methods
Line 99 define TCBS and TSA
Line 95 “Samples with curcumin were prepared by adding 5 ml of fresh V. parahaemolyticus, 95 and A. hydrophila cell suspensions to 5 ml of curcumin solution (1, 5, and 10 mg/l)”. In this conditions the final concentrations of curcumin should be 0.5, 2.5 and 5 mg/l, please check these values in figures or change (1, 5 and 10 mg/l) by (2, 10 and 20 mg/l).
Line 96 “Samples were placed into a sterile 6-well clear polystyrene microplates and were treated by UV-A for 5, and 15 min, and LED for 5, 15, and 30 min at 4 and 22 °C” Explain the differences of treatment times, why for LED is longer
Line 118-119 “9 ml of nanobubbles” change by “9 ml of solution where nanobubbles were produced”
- Results and discussion
Line 131 “between 1 and 10 mg/ml inactivated 2 to more than 5 log cfu/ml V. parahaemolyticus” ¿respectively?
Line 132 “4.5 cfu/ml” change by “4.5 log cfu/ml”
- The caption of figures repeat information showed in the figures, the limit of detection and the control samples used should be explained in Materials and methods and the results of control samples should be explained in Results and discussion not in the caption.
Recommendation: Figure 4 is represented as bacterial reduction (y axis), and the other figures as bacterial count, it is better for readers the same way for all figures in the paper.
- Conclusion
Line 266 “UV-A and visible LED in combination with food-grade curcumin can rapidly inactivate aquatic pathogens in water. High bacterial inactivation using curcumin at room and refrigerator temperatures, offering new possible sanitation approach for seafood sanitation” Remove one sanitation
References
Line 281 check the correct way of book reference according to journal format.
Author 1, A.; Author 2, B. Title of the chapter. In Book Title, 2nd ed.; Editor 1, A., Editor 2, B., Eds.; Publisher: Publisher Location, Country, Year; Volume 3, pp. 154–196.
2- Kumar, S.; Lekshmi, M.; Parvathi, A.; Nayak, B.B.; Varela, M.F. Antibiotic Resistance in Seafood Borne Pathogens. In Food Borne Pathogens and Antibiotic Resistance; Singh, O.V., Ed.; John Wiley & Sons: Hoboken, NJ, USA, 2017, pp. 397-416.
Also for articles, the references should be checked
- Author 1, A.B.; Author 2, C.D. Title of the article. Abbreviated Journal Name Year, Volume, page range.
Author Response
Thank you for useful and helpful comments. We have revised the manuscript based on your comments, and applied all of them.
Reviewer 2 Report
This manuscript reports on the antibacterial effectiveness of photosensitizer (activated curcumin via exposure to LED and/or UV-A) and nanobubbles with ultrasound against V. parahaemolyticus and A. hydrophila. This research is important because it illustrates potential use of cucurmin activated by light technologies for improving the microbial quality of water in aquaponics systems while reducing dependency on chemicals. Generally, the manuscript is well written with clear objectives and methods that are appropriate for answering the research questions. The results are adequately interpreted with reference to pertinent published research and the conclusions are justifiable based on the results. Please see the following comments and suggestions for improving the manuscript.
Line 30: "..are growing industry". Insert "a" in front of "growing"
Line 47: "..application of photodynamic have emerged..". This part of the sentence is unclear. The word "photodynamic" is not a noun; also there is incorrect subject and verb agreement.
Line 48: Typographical error: "photosensitizes"
Line 56: "cause cavitation on cell microbial cell wall". The word "cell" stated before "microbial" is unnecessary.
Line 64: Write the full term for the abbreviation "RAS" if this is the first time the term is being used in the manuscript.
Line 81: "1 ml": Avoid placing a numeral at the beginning of a sentence; write the full word eg "One".
Line 82: "12,000 rpm". Please write the relative centrifugal force ( .... x g) in place of the rpm so that researchers with a different centrifuge rotor could accurately repeat the method. State the temperature at which the centrifugation was performed. Please make this correction in other areas of the manuscript where appropriate.
Line 83: "pellets were resuspended". The pellets are clumps of cells resulting from centrifugation of suspended cells and those clumps were not actually suspended. Suggestion: " the pelleted cells were resuspended..."
Line 105: "Prior conducting the experiment". Insert "to" before "conducting".
Line 118: "1 ml": Avoid placing a numeral at the beginning of a sentence; write the full word eg "One".
Line 139" Insert "for" before "which"
Lines 259-262. This sentence is unclear; please rewrite it to improve its clarity.
Author Response
Dear reviewer, thank you for providing us these useful comments. We have applied all of them and revised the MS based on your comments.
Reviewer 3 Report
Major Comment
- Subsection 2.6: Authors mention that they used triplicate experiments and comparisons - significance of differences were determined using ANOVA. Authors need to mention if and with what method they checked normality of their Data in order to be able to use ANOVA in such a small number of repetitions.
Minor Comments
- Line 36: needs rephrasing
- Line 39: the word “this” needs to be replaced by the word “the”
- Line 88: Why these concentrations of curcumin were chosen? Are they relevant to dose safety of curcumin in fish during aquaculture?
- In line 96 authors mention that curcumin concentration was 1, 5 and 10 mg/L in experiments, while in line 131 they mention a concentration of 1 to 10 mg/ml of curcumin. This needs to be clarified and corrected consistently throughout the document.
- Lines 138-140 and 196-198 need rephrasing (i.e. the word “in” can be added before the word “which”).
- Figures 1 and 2: The authors need to indicate with appropriate symbols the statistical differences between the samples within the figures for different concentrations of curcumin and different temperatures. For example, it is better to highlight the statistical significant differences with the asterisk (*) and with another symbol the “below the limit of detection” samples that was used in Figure 1.
- Figure 3: The authors need to indicate with appropriate symbols the statistical differences between the samples within the figures for different time points and aquaponics/control. For example, it is better to highlight the statistical significant differences with the asterisk (*) and with another symbol the “below the limit of detection” samples that was used in Figure 3a.
- Similar comment for Figure 4.
- Why did the authors chose to use bacterial reduction in Y-axis in Figure 4 and not bacterial count, as they did in the other figures? This needs to be clarified and a relevant phrase needs to be used within the text.
Author Response
Thank you for providing us these useful comments.
All the comments were applied and we revised the MS based on them.
Also, please note that, all the experiments were repeated at least two times (n=6), which we have revised it.
Round 2
Reviewer 3 Report
Major Revision
- The suthors still need to state at the statistics section of the Methods Section the test they used for checking the normality of their data, and they need to mention to their results, the outcome of this testing. Otherwise the use of the ANOVA test for comparisons cannot be justified.
- The authors have not mentioned anything about the safety of the concentrations of curcumin that they tested. Have these concentrations applied safely in aquaculture systems by the authors or by other studies, without any side effect to marine organisms? If yes, this needs to be added and mentioned to their text
Minor Comments
- In line 90 and in Figure 1 authors mention that curcumin concentration was 1, 5 and 10 mg/L in experiments, while in line 138 the authors still use the concentration of curcumin as mg/ml and not as mg/L. This still needs to be corrected and this correction needs to be performed consistently throughout the document.
- In legends of both Figure 3 and 4 authors need to state what the two asterisks (**) stands for
Author Response
Major Revision
- The authors still need to state at the statistics section of the Methods Section the test they used for checking the normality of their data, and they need to mention to their results, the outcome of this testing. Otherwise the use of the ANOVA test for comparisons cannot be justified.
- Response: Thank you. We used KS for normality, and we added the software name to the MS as well.
- The authors have not mentioned anything about the safety of the concentrations of curcumin that they tested. Have these concentrations applied safely in aquaculture systems by the authors or by other studies, without any side effect to marine organisms? If yes, this needs to be added and mentioned to their text
- Response: Thank you for the important point. According to the literature, exposing fish up to 10 days in the tested range of curcumin is safe. The LC50 is around 60 mg/l which is 6 times greater that our tested concentrations. We brought one reference (39) into the MS to confirm it.
Minor Comments
- In line 90 and in Figure 1 authors mention that curcumin concentration was 1, 5 and 10 mg/L in experiments, while in line 138 the authors still use the concentration of curcumin as mg/ml and not as mg/L. This still needs to be corrected and this correction needs to be performed consistently throughout the document.
- Response: Thank you for the comment, it has been revised to mg/l.
- In legends of both Figure 3 and 4 authors need to state what the two asterisks (**) stands for
- Response: Thank you for the comment, it has been revised.